# Mental Health Evaluation in Community Pharmacies—A Cross-Sectional Study

**DOI:** 10.3390/pharmacy12030089

**Published:** 2024-06-07

**Authors:** Mónica Condinho, Isabel Ramalhinho, Catarina Vaz-Velho, Carlos Sinogas

**Affiliations:** 1AcF—Acompanhamento Farmacoterapêutico, Lda., 7490-324 Pavia, Portugal; sinogas@sapo.pt; 2Faculty of Sciences and Technology, University of Algarve, 8005-139 Faro, Portugal; iramalhinho@ualg.pt; 3ABC-RI, Algarve Biomedical Center Research Institute, 8005-139 Faro, Portugal; 4Department of Psychology, University of Évora, 7004-516 Évora, Portugal; vazvelho@uevora.pt; 5Department of Medical and Health Sciences, University of Évora, 7004-516 Évora, Portugal

**Keywords:** mental health, pharmacist, community pharmacy

## Abstract

Portugal has a high prevalence of anxiety and depression, and community pharmacists are well-placed to identify mental health problems and monitor their treatment. This study aimed to screen undiagnosed people for symptoms of anxiety and depression and to monitor these conditions in diagnosed patients. We conducted an observational, cross-sectional study of a sample of community pharmacy users. Anxiety and depression symptoms were assessed using the Generalised Anxiety Disorder-7 (GAD-7) and Patient Health Questionnaire-9 (PHQ-9). Altogether, 591 participants were enrolled between September 2020 and July 2021, 74.9% of whom were female. Of the 477 undiagnosed participants who completed the GAD-7, 87 (18.2%) had moderate or severe anxiety symptoms. Of the 485 participants who completed the PHQ-9, 59 (12.1%) had moderate or severe symptoms of depression. Of the 94 patients diagnosed with anxiety, 37 (39.4%) reported moderate to severe symptoms. Similarly, of the 97 patients with depression, almost half (49.5%) reported moderate to severe symptoms. Anxiety levels were lower among men and among those who reported not taking any medication (*p* < 0.001). Moderate or severe symptoms of depression were more common among women (*p* < 0.001), participants with lower education levels (*p* < 0.005), participants who live alone (*p* < 0.007), and those taking medication for insomnia (*p* < 0.001), pain (*p* < 0.001), anxiety (*p* < 0.001), and/or depression (*p* < 0.001). Screening for anxiety and depression revealed that a significant proportion of undiagnosed participants had clinically relevant symptoms of anxiety and depression. However, among participants diagnosed with anxiety and depression, between 40% and 50% were uncontrolled, respectively. These data highlight pharmacists’ role in identifying customers at risk of anxiety and depression as well as the need for better monitoring of those already diagnosed.

## 1. Introduction

Maintaining solid mental health is essential for a healthy and productive life [1]. Portugal has the highest burden of mental illness in the European Union (EU) [2]. According to prevalence estimates from the Institute for Health Metrics and Evaluation (IHME), more than 2.25 million people in Portugal had a mental health disorder in 2019, representing 22% of the population—higher than the EU average of 16.7%. Anxiety disorders were the most common, affecting around 9% of the population, followed by depressive disorders, at 6% [2]. It is estimated that in 2019, mental health disorders resulted in the loss of nearly 310,000 years of productive life [3]. Furthermore, the direct and indirect costs associated with mental health problems in Portugal (2015) were estimated to be around 3.7% of GDP [4]. The COVID-19 pandemic (2020–2023) caused a significant increase in anxiety and depression in all EU countries [5,6], and Portugal was no exception [7]. 

Despite National Health Service efforts, the low supply of mental health professionals and regional disparities make it difficult for people to access services [2].

Therefore, it is essential that all health professionals help reduce the burden of mental illness, particularly anxiety and depression [8]. Early detection of these mental disorders through screening is recommended, as is monitoring the effectiveness and safety of treatment in people who already have been diagnosed and treated [9]. According to the International Pharmaceutical Federation (FIP), pharmacists can promote their patients’ mental health and well-being to help prevent people from developing mental illness. They also can use their accessibility to identify patients who may be experiencing a mental health problem by being able to recognise symptoms of mental illness or conducting screenings to determine whether a patient needs to be referred for further assessment or follow-up care [10]. Pharmacists also can help monitor people who already have been diagnosed [10]. However, from a practical perspective, pharmacists must be aware of their intervention’s potential impact on the mental health of people whom they interact with on a daily basis [11].

This study aimed to screen undiagnosed people for symptoms of anxiety and depression and to monitor these conditions in diagnosed patients by community pharmacists. Furthermore, we aimed to assess symptoms’ prevalence based on sociodemographic factors, chronic illness, and isolation.

## 2. Materials and Methods

An observational and cross-sectional study was designed. A pharmacy team (pharmacists and pharmacy technicians) selected the sample, comprising users from different community pharmacies in the southern region of the country using a non-probabilistic method during daily activities (a real-world study). Under this method, participants who met the inclusion criteria were selected based on the pharmacy team and costumer’s availability to participate. Inclusion criteria were an age of 18 or older, being able to provide informed consent and fill out a questionnaire, and having the cognitive ability to understand the study’s purpose. People with a diagnosis of Alzheimer’s disease were excluded from the study, as were people who were seen outside the pharmacy’s normal opening hours (e.g., emergency services).

The participating pharmacies were those that accepted the research team’s invitation. Pharmacies were invited according to their geographical distribution in terms of accessibility to the research team. The majority of the pharmacies were located in urban areas.

Between September 2020 and July 2021, data were collected using a self-administered questionnaire specifically designed for this purpose. The questionnaire sought data on sociodemographic attributes, the existence of pets at home, and health problems, as well as data from self-administered scales to assess the presence of anxiety and depression symptoms, namely the Generalised Anxiety Disorder-7 (GAD-7) [12,13] and Patient Health Questionnaire-9 (PHQ-9) [14,15]. 

The GAD-7 comprises seven questions, each of which has four options, from which the participant should select only one. Each option is scored from 0 to 3, with 0 assigned to “never” and 3 to “almost every day”. The rating is based on the score obtained by summing up the question scores, which are translated qualitatively into mild, moderate, and severe anxiety, i.e., ≥5, ≥10, and ≥15, respectively [12] Anxiety is viewed as clinically significant for scores ≥ 10 [12]. The PHQ-9 comprises nine questions, in which the person must select one of the four options presented. These are rated from 0 (“never”) to 3 (“almost every day”), and the final score is translated qualitatively into mild, moderate, moderately severe, and severe depression, i.e., ≥5, ≥10, ≥15, and ≥20, respectively [14]. Depression is viewed as clinically relevant for scores ≥ 10 [16].

Once the target customers were identified, the pharmacy employee invited them to take part in the study by briefly explaining the objectives. If they accepted, they signed the informed consent form and filled in the questionnaire, which was completed at the pharmacy in an area with greater privacy. Once completed, it was returned to the pharmacy team and filed for later delivery to the research team. 

The main outcome measures comprised the scores on the anxiety and depression scales.

A pilot study was also conducted to assess the questionnaire’s applicability and suitability to the intended objectives. The pilot study also made it possible to reduce information and classification bias.

The data were processed and analysed using SPSS (Version 28) statistical software. After a descriptive analysis of the sample characteristics, the Mann–Whitney and Kruskal–Wallis tests were used for bivariate analysis, as normality assumptions were not met. The Kolmogorov–Smirnov/Shapiro–Wilk test and the Levene test were used to test for normality in data distribution and homogeneity of variances, respectively. The significance level considered was 0.05, and the confidence interval was 95%. Mean values were presented as mean and standard deviation (SD). Missing data were eliminated and not used for the percentages, as indicated in the results tables.

The study was approved by the ethics committee of the University of Algarve (CEUAlg Pn°32a)/2021). All research was conducted in accordance with relevant guidelines and regulations.

## 3. Results

Altogether, 591 people participated in the study (28 pharmacies), most of whom (n = 441; 74.9%) were females. The mean age was 50.2 years (SD = 16.6). Secondary education (33.2%) and higher education (33.2%) were the most prevalent in the sample. Of those surveyed, 401 (68.1%) were employed. Most of the participants (501; 85.9%) reported living with someone, and 340 had pets (59.1%). The sample’s sociodemographic characteristics are presented in Table 1.

Almost two-thirds of the sample (372; 62.9%) said they took medication for some health problem. The most prevalent health problems were hypertension (26.2%), hypercholesterolaemia (18.6%), and anxiety (17.3%), followed by depression (16.6%), pain (15.6%), insomnia (11.8%), diabetes (7.6%), and asthma (5.6%).

### 3.1. GAD-7 and PHQ-9 Global Scores

Analysing the results from the anxiety and depression scales globally, we obtained an average score for the GAD-7 (n = 571) of 6.4 (SD = 4.8), and for the PHQ-9 (n = 582) of 5.6 (SD = 5.1). Based on pathology, those who reported taking medication for insomnia (n = 70; GAD-7 = 10.8; PHQ-9 = 10.7), depression (n = 98; GAD-7 = 10.2; PHQ-9 = 10.6), and anxiety (n = 102; GAD-7 = 9.5; PHQ-9 = 9.7) had the highest average scores. 

Segregating the results based on the detected symptoms’ intensity, of the 571 participants who completed the GAD-7, 229 (40.1%) had mild symptoms; 74 (12.9%), moderate; and 47 (8.2%), severe. As for the PHQ-9, of the 582 responses, 160 (27.5%) reported mild symptoms; 59 (10.1%), moderate; and 49 (8.4%), severe (note: For data processing purposes, the “severe” category included people who had scores compatible with “moderately severe depression” and “severe depression”).

### 3.2. GAD-7 and PHQ-9 Scores in Undiagnosed and Diagnosed Participants with Anxiety or Depression

Regarding the assessment of anxiety symptoms, of the participants who filled in the GAD-7 (n = 571), 94 (16.5%) reported taking medication for anxiety, i.e., they were assumed to have a diagnosis of anxiety. Of the 477 (83.5%) who denied taking these drugs, 183 (38.4%) had mild symptoms of anxiety; 62 (13.0%), moderate; and 25 (5.2%), severe. Among those diagnosed with anxiety, 43 (45.7%) had mild symptoms; 15 (16.0%), moderate; and 22 (23.4%), severe (Figure 1).

Regarding assessment of depression symptoms, 582 completed the PHQ-9. Of these, 97 (16.7%) reported taking medication for depression, so it was assumed that they had received a depression diagnosis. The remaining 485 (83.3%) denied taking medication for this condition. In the first case, despite being treated, most of the participants had moderate (19.6%) and severe (29.9%) symptoms of depression. Furthermore, 29 (29.9%) had mild symptoms. In the second case, 133 (27.4%) reported mild symptoms; 40 (8.2%), moderate; and 19 (3.9%), severe (Figure 1).

### 3.3. Analysis of Anxiety and Depression Symptoms’ Frequency

Based on the bivariate analysis of our data, the occurrence of anxiety and depression symptoms is related to sex. Female sex is associated with the development of more severe anxiety and depression symptoms (*p* < 0.001). However, age does not seem to influence the development of anxiety and depression symptoms (*p* > 0.05) (Table 2 and Table 3).

Education also seems to be a relevant factor in the development of depression symptoms—specifically, a higher level of education appears to be a “protective factor” in relation to the development of depression symptoms (Table 3). Statistically significant differences were found between the group with the lowest education level and those with the highest education level: secondary (KW = 51.148; *p* = 0.008) and higher (KW = 63.434; *p* < 0.001).

Although not statistically significant, a tendency appears to exist between being unemployed and having a higher score for symptoms of depression (*p* > 0.05) (Table 3).

No statistically significant association was found between living alone and anxiety symptoms (*p* = 0.190), but in the case of depression, it was found that living alone or with someone exerted a statistically significant effect on the development of depression symptoms (*p* = 0.007) (Table 2 and Table 3).

The presence of pets exerted no influence on anxiety and depression symptoms (*p* = 0.347 and *p* = 0.219, respectively).

Regarding medication use, taking medication for insomnia (*p* < 0.001), pain (*p* = 0.005 and *p* < 0.001, respectively), anxiety (*p* < 0.001), and/or depression (*p* < 0.001 and *p* < 0.000, respectively) was associated favourably with the development of moderate and severe anxiety and depression symptoms (Table 4 and Table 5). However, no statistical association (*p* > 0.05) was found between taking medication for other chronic diseases (e.g., arterial hypertension, hypercholesterolaemia, diabetes, and asthma) and anxiety and depression symptoms.

## 4. Discussion

Our results were derived from a non-probabilistic sample that was not representative of the Portuguese population, but they are of relevant clinical interest because they comprise real-life data [17], reflecting professional practice in community pharmacies in both urban and rural areas in the nation’s southern region.

Our sample contains characteristics that diverge from the southern region’s population, namely the prevalence of women (higher in our sample) [18], education level (higher in our sample) [19], the percentage of people living alone (higher in our sample) [19], and the prevalence of people with anxiety and depression (higher in our sample) [20]. Along with choosing a non-probabilistic sample, the sample’s aspects can be attributed to women’s tendency to be more likely to participate in health promotion activities [21] and to people with higher education levels to be more likely to feel comfortable filling out questionnaires [22,23]. Similarly, people who live alone and those with mental health problems tend to feel more in need of care; therefore, they are more likely to participate in health activities.

In this context, the study reveals the reality of the people who patronise community pharmacies, which adds to the value associated with data from a reality that has not been studied widely, in addition to the value associated with real-life data [17], as most anxiety and depression studies are conducted in primary healthcare units or hospitals [24,25,26,27].

To assess anxiety and depression, the GAD-7 and PHQ-9 were chosen over other scales, including the Depression and Anxiety Stress Scale (DASS) [28], the Hamilton Rating Scale for Anxiety (HAMA) [29], and the Hospital Anxiety and Depression Scale [30]. Incidentally, the HAMA scale is the gold standard for assessing anxiety symptoms’ severity, but it has not been validated in the Portuguese population from Portugal, so it should be used cautiously [31]. Furthermore, its use is recommended in the context of a structured interview [32], which was not the case in our study. Thus, the choice of GAD-7 and PHQ-9 is justified because both have been validated for clinical use in Portugal [13,15] and can be used with people who have suspected generalised anxiety disorder/depression, as well as with people who already have been diagnosed, to monitor symptoms and adjust treatment [15,33].

Regarding the results of the application of the anxiety and depression scales, our results overall indicate that 21.1% and 18.5% of the sample had moderate to severe anxiety and depression symptoms, respectively. These values are lower than those found in a study conducted in the northern area of Portugal, where the prevalence of moderate to severe symptoms reached 35.5% for anxiety and 20% for depression [34]. This variance may be attributed to the other study being conducted using a different tool for assessing anxiety and depression, DASS-21. Moreover, it was conducted in a health unit [34], i.e., not in a community pharmacy, which in itself could influence the data. Understandably, people with health problems tend to go to health centres instead of pharmacies, where the population tends to be more heterogeneous in terms of health status.

When comparing our results with the prevalence of generalised anxiety and depression in Sweden, our estimates are higher [35]. Despite using the same tools to assess anxiety and depression (GAD-7 and PHQ-9), that study estimated a prevalence of 14.7% for clinically relevant generalised anxiety disorder and 10.8% for depression [35].

A study conducted in Germany in 2021 on a representative sample of the population (18–70 years old) pointed to a prevalence of anxiety (GAD-7 score ≥ 10) that also was lower (13.4%) than our results; however, the prevalence of depression (PHQ-9 score ≥ 10) proved to be less disparate (20.0%) from our data. These differences can be explained by the German study’s design, which involved an online survey of participants who were younger on average (average age 44.5) than those in our sample [36].

Therefore, we assert that the different prevalence estimates of moderate to severe (clinically relevant) anxiety and depression symptoms between our study and the literature may be tied to differences in the target populations, study designs, and settings. 

In analysing the mean scores for anxiety and depression in our study, we found values of 6.4 and 5.6, respectively. These values are higher than others found in the literature (between 3.59 [35] and 4.32 [37] for anxiety and 3.70 [35] for depression) and lower than others (8.1 for anxiety and 11.2 for depression [38]), which may be explained by the differences in study designs.

As for the positive relationship between anxiety and depression and female sex, as in other published studies [35,39], our results point in the same direction: the female sex is associated with more severe anxiety and depression symptoms (*p* < 0.001).

Notably, our study found that people with pain had higher levels of generalised anxiety (*p* = 0.035) and depression (*p* < 0.001), while the literature indicate that people with pain tend to have higher levels of depression and anxiety [40,41], which COVID-19 aggravated [41].

### 4.1. Screening for Anxiety and Depression (Participants without a Diagnosis of Anxiety or Depression)

According to our study, among people without a diagnosis of anxiety who completed the GAD-7 (n = 477), 18.2% had moderate to severe symptoms (score ≥ 10). Among those without a diagnosis of depression who completed the PHQ-9 (n = 485), 12.1% also had moderate to severe symptoms (score ≥ 10). 

A study conducted in 15 primary healthcare clinics in the United States (n = 965) to determine anxiety disorder prevalence found a prevalence of generalised anxiety disorder of 7.6% [24], a figure much lower than that found in our study.

In Europe, another study conducted in Spain in primary healthcare units (n = 260), which aimed to screen participants for anxiety and depression symptoms using the GAD-7 and PHQ-9, found levels of moderate to severe anxiety and depression much higher than ours: 69% and 78%, respectively [26]. However, the authors acknowledge that the study’s limitations may have influenced the results, namely that the questionnaire was administered on a computer (which may have made it difficult to interpret the questions), the population was very heterogeneous, and the sample size was small. Furthermore, the study yielded a large number of false positives [26]. In Finland, generalised anxiety disorder’s prevalence among frequent users of health care [eight or more visits per year to the general practitioner (GP) or four or more visits to the hospital] (n = 150) was estimated at 4%, a value that the study’s authors viewed as low [27].

Most of the studies found in the literature were conducted in primary healthcare units and not in community pharmacies [24,25,26,27], which also may help explain the discrepancy found in both the literature and our data. Furthermore, methodological differences limited the comparability of results.

However, despite the inconsistencies found between studies, in line with two recent similar studies [38,42], our results suggest that pharmacists need to be involved in screening for anxiety and depression to identify these health problems early, refer them to a physician for diagnosis and treatment, and monitor outcomes.

### 4.2. Assessment of Anxiety and Depression (Participants Diagnosed with Anxiety and Depression)

In our sample, of those diagnosed with anxiety who completed the GAD-7 (n = 94), 39.4% had moderate to severe symptoms, indicating a lack of control over the condition. Of those diagnosed with depression who completed the PHQ-9 (n = 97), 49.5% also had moderate to severe symptoms, again indicating that almost half the sample subjects were not in control of their condition despite taking medication. 

These data highlight the need for strategies to monitor prescribed therapy to maximise its effectiveness. A study (n = 50) evaluating the impact of integrating a clinical pharmacist into the healthcare team in primary health care found that this strategy led to a statistically and clinically significant improvement in scores (GAD-7 and PHQ-9) for various mental health disorders by supporting medication management, particularly by promoting adherence to therapy [43]. Another study published in 2019 described a 12-month pilot trial conducted in rural Scotland, in which patients (n = 75) were referred by their GP to an independent prescribing pharmacist specialising in mental health [44]. Of the patients involved, 34 reduced their PHQ-9 and GAD-7 scores by 50%. The authors concluded that the mental health prescribing pharmacist provided a high quality service to people diagnosed with moderate to severe anxiety and depression and that it was a model of care that could be implemented in general practice [44].

Other studies also have demonstrated that pharmacists play an important role in improving the care of people with anxiety and depression [38,42,45,46,47,48,49], particularly when they are part of a healthcare team [45,47,49]. E-consultation also appears to have a positive impact in less complicated clinical situations [50].

Given the high proportion of participants we identified with clinically relevant symptoms of anxiety and depression (whether or not they had a pre-existing diagnosis of anxiety and depression), we believe that more research is needed on the role of community pharmacists in anxiety and depression, particularly randomised controlled trials. In line with other published studies, our findings support the design and implementation of pharmacist-led clinical services for generalised anxiety disorder and depression.

## 5. Conclusions

Screening for anxiety and depression revealed that a significant proportion of undiagnosed participants had clinically relevant symptoms (GAD-7 and/or PHQ-9 ≥ 10) of anxiety (~18%) and depression (~12%). 

However, among participants diagnosed with anxiety and depression, approximately 40% and 50% of them, respectively, were uncontrolled (GAD-7 and/or PHQ-9 ≥ 10).

The presence of clinically relevant symptoms of anxiety and depression was associated favourably with being female and taking medication for insomnia, pain, anxiety and depression.

In this context, our results highlight the pharmacist’s role in identifying people at risk of developing generalised anxiety disorder and depression by using questionnaires to assess characteristic symptoms. Furthermore, for people who already have been diagnosed, a clear need exists to monitor the effectiveness of therapy through clinical services developed by pharmacists.

## Figures and Tables

**Figure 1 pharmacy-12-00089-f001:**
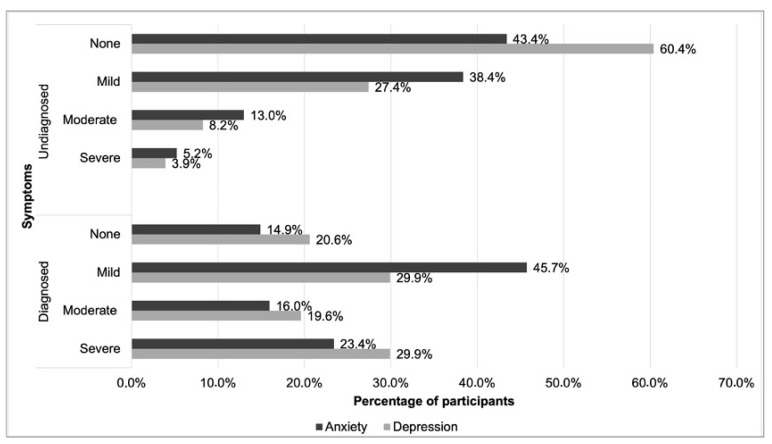
GAD-7 and PHQ-9 scores for undiagnosed and diagnosed participants with anxiety and depression, respectively.

**Table 1 pharmacy-12-00089-t001:** The sample’s sociodemographic characteristics.

Sociodemographic Characteristics	Frequency (N)	Percentage (%)
Sex		
Male	148	25.1
Female	441	74.9
NA	2	-
Age		
18–34	114	21.1
35–64	306	56.7
≥65	120	22.2
NA	51	-
Education		
No school	5	0.9
1st cycle (4 years)	89	15.3
2nd cycle (6 years)	30	5.2
3rd cycle (9 years)	72	12.4
Secondary (12 years)	193	33.2
Higher	193	33.2
NA	9	-
Professional status		
Employee *	401	68.1
Retired	132	22.4
Unemployed	56	9.5
NA	2	-
Live alone		
Yes	82	14.1
No	501	85.9
NA	8	-
Pets		
Yes	340	59.1
No	235	40.9
NA	16	-

* In this category, we included people working (384) and people on sick leave (17). NA—Not applicable (missing).

**Table 2 pharmacy-12-00089-t002:** Symptoms of anxiety (GAD-7) vs. sociodemographic characteristics.

	GAD-7 (Qualitative Scores)						
Sociodemographic Characteristics	No Symptoms N (%)	Mild N (%)	Moderate N (%)	Severe N (%)	Total N (%)	Mean (CI 95%) Median	Significance
Sex							U = 40,042.000; *p* < 0.001
Male	86 (58.9)	42 (28.8)	13 (8.9)	5 (3.4)	146 (100)	0.57 (0.44; 0.70) 0.00	
Female	133 (31.4)	184 (43.5)	64 (15.1)	42 (9.9)	423 (100)	1.04 (0.95; 1.12) 1.00	
Age							KW = 1.415; *p* = 0.493
18–34	45 (39.5)	50 (43.9)	13 (11.4)	6 (5.3)	114 (100)	0.82 (0.67; 0.98) 1.00	
35–64	107 (36.1)	120 (40.5)	41 (13.9)	28 (9.5)	296 (100)	0.97 (0.86; 1.07) 1.00	
65–83	48 (42.5)	35 (31.0)	18 (15.9)	12 (10.6)	113 (100)	0.95 (0.76; 1.13) 1.00	
Education level							KW = 4.443; *p* = 0.217
<Middle school	29 (33.3)	32 (36.8)	13 (14.9)	13 (14.9)	87 (100)	1.11 (0.89; 1.34) 1.00	
Middle school	41 (41.8)	30 (30.6)	15 (15.3)	12 (12.2)	98 (100)	0.98 (0.77; 1.19) 1.00	
Secondary	78 (41.3)	69 (36.5)	29 (15.3)	13 (6.9)	189 (100)	0.88 (0.75; 1.01) 1.00	
Higher	72 (38.3)	89 (47.3)	18 (9.6)	9 (4.8)	188 (100)	0.81 (0.69; 0.92) 1.00	
Professional status							KW = 0.769; *p* = 0.681
Active/Medical leave	153 (39.1)	162 (41.4)	44 (11.3)	32 (8.2)	391 (100)	0.88 (0.79; 0.97) 1.00	
Retired	50 (40.0)	42 (33.6)	21 (16.8)	12 (9.6)	125 (100)	0.96 (0.79; 1.13) 1.00	
Unemployed	18 (34.0)	22 (41.5)	11 (20.8)	2 (3.8)	53 (100)	0.94 (0.71; 1.18) 1.00	
With whom do you live?							U = 17,315.500; *p* = 0.190
Live alone	27 (34.6)	29 (37.2)	12 (15.4)	10 (12.8)	78 (100)	1.06 (0.84; 1.29) 1.00	
Live together	190 (39.1)	195 (40.1)	65 (13.4)	36 (7.4)	486 (100)	0.89 (0.81; 0.97) 1.00	
Pet owners							U = 35,711.500; *p* = 0.347
Yes	126 (38.0)	130 (39.2)	45 (13.6)	31 (9.3)	332 (100)	0.94 (0.84; 1.04) 1.00	
No	90 (40.0)	92 (40.9)	30 (13.3)	13 (5.8)	225 (100)	0.85 (0.74; 0.96) 1.00	

**Table 3 pharmacy-12-00089-t003:** Symptoms of depression (PHQ-9) vs. sociodemographic characteristics.

	PHQ-9 (Qualitative Scores)						
Sociodemographic Characteristics	No Symptoms N (%)	Mild N (%)	Moderate N (%)	Severe N (%)	Total N (%)	Mean (CI 95%) Median	Significance
Sex							U = 39,394.500; *p* < 0.001
Male	105 (70.9)	25 (16.9)	12 (8.1)	6 (4.1)	148 (100)	0.45 (0.32; 0.58) 0.00	
Female	206 (47.7)	137 (31.7)	47 (10.9)	42 (9.7)	432 (100)	0.83 (0.73; 0.92) 1.00	
Age							KW = 0.0722; *p* > 0.05
18–34	60 (53.1)	34 (30.1)	11 (9.7)	8 (7.1)	113 (100)	0.71 (0.54; 0.88) 0.00	
35–64	158 (52.1)	91 (30.0)	33 (10.9)	21 (6.9)	303 (100)	0.73 (0.62; 0.83) 0.00	
65–83	61 (52.6)	25 (21.6)	13 (11.2)	17 (14.7)	116 (100)	0.88 (0.68; 1.08) 0.00	
Education level							KW = 13.031 *p* < 0.005
<Middle school	38 (42.2)	26 (28.9)	12 (13.3)	14 (15.6)	90 (100)	1.02 (0.79; 1.25) 1.00	
Middle school	51 (50.5)	24 (23.8)	14 (13.9)	12 (11.9)	101 (100)	0.87 (0.66; 1.08) 0.00	
Secondary	107 (56.3)	53 (27.9)	15 (7.9)	15 (7.9)	190 (100)	0.67 (0.54; 0.81) 0.00	
Higher	113 (58.9)	56 (29.2)	16 (8.3)	7 (3.6)	192 (100)	0.57 (0.45; 0.68) 0.00	
Professional status							KW = 4.950 *p* = 0.084
Active/Medical leave	221 (55.8)	116 (29.3)	33 (8.3)	26 (6.6)	396 (100)	0.66(0.57; 0.74) 0.00	
Retired	66 (51.2)	35 (27.1)	14 (10.9)	14 (10.9)	129 (100)	0.81(0.64; 0.99) 0.00	
Unemployed	26 (47.3)	11 (20.0)	11 (20.0)	7 (12.7)	55 (100)	0.98(0.69; 1.28) 1.00	
With whom do you live?							U = 16,227.500 *p* = 0.007
Live alone	35 (44.3)	18 (22.8)	13 (16.5)	13 (16.5)	79 (100)	1.05 (0.80; 1.3) 1.00	
Live together	273 (55.2)	142 (28.7)	46 (9.3)	34 (6.9)	495 (100)	0.68 (0.60; 0.76) 0.00	
Pet owners							U = 36,612.000; *p* = 0.219
Yes	174 (52.1)	94 (28.1)	34 (10.2)	32 (9.6)	334 (100)	0.77 (0.67; 0.88) 0.00	
No	131 (56.5)	63 (27.2)	24 (10.3)	14 (6.0)	232 (100)	0.66 (0.54; 0.77) 0.00	

**Table 4 pharmacy-12-00089-t004:** Symptoms of anxiety (GAD-7) vs. taking medicine.

	GAD-7 (Qualitative Scores)						
Characteristics	No Symptoms N (%)	Mild N (%)	Moderate N (%)	Severe N (%)	Total N (%)	Mean (CI 95%) Median	Significance
Taking medicine							U = 30,901.000; *p* < 0.001
Yes	121 (34.1)	136 (38.3)	59 (16.6)	39 (11.0)	355 (100)	1.05 (0.94; 1.15) 1.00	
No	100 (46.3)	90 (41.7)	18 (8.3)	8 (3.7)	216 (100)	0.69 (0.59; 0.80) 1.00	
Pain medicine							U = 17,186.000; *p* = 0.005
Yes	24 (27.9)	34 (39.5)	18 (20.9)	10 (11.6)	86 (100)	1.16 (0.96; 1.37) 1.00	
No	197 (40.6)	192 (39.6)	59 (12.2)	37 (7.6)	485 (100)	0.87 (0.79; 0.95) 1.00	
Insomnia medicine							U = 8266.000; *p* < 0.001
Yes	7 (10.9)	22 (34.4)	17 (26.6)	18 (28.1)	64 (100)	1.72 (1.47; 1.97) 2.00	
No	214 (42.2)	204 (40.2)	60 (11.8)	29 (5.7)	507 (100)	0.81 (0.74; 0.89) 1.00	
Anxiety medicine							U = 14,019.500; *p* < 0.001
Yes	14 (14.9)	43 (45.7)	15 (16.0)	22 (23.4)	94 (100)	1.48 (1.27; 1.69) 1.00	
No	207 (43.4)	183 (38.4)	62 (13.0)	25 (5.2)	477 (100)	0.80 (0.72; 0.88) 1.00	
Depression medicine							U = 11,474.000; *p* < 0.001
Yes	10 (10.9)	36 (39.1)	24 (26.1)	22 (23.9)	92 (100)	1.63 (1.43; 1.83) 1.50	
No	211 (44.1)	190 (39.7)	53 (11.1)	25 (5.2)	479 (100)	0.77 (0.70; 0.85) 1.00	

**Table 5 pharmacy-12-00089-t005:** Symptoms of depression (PHQ-9) vs. taking medicines.

	PHQ-9 (Qualitative Scores)						
Characteristics	No Symptoms N (%)	Mild N (%)	Moderate N (%)	Severe N (%)	Total N (%)	Mean (CI 95%) Median	Significance
Taking medicine							U = 30,976.500; *p* < 0.001
Yes	174 (47.7)	100 (27.4)	48 (13.2)	43 (11.8)	365 (100)	0.89 (0.78; 1.00) 1.00	
No	139 (64.1)	62 (28.6)	11 (5.1)	5 (2.3)	217 (100)	0.46 (0.36; 0.55) 0.00	
Pain medicine							U = 138,898.000; *p* < 0.001
Yes	35 (39.3)	24 (27.0)	18 (20.2)	12 (13.5)	89 (100)	1.08 (0.85; 1.30) 1.00	
No	278 (56.4)	138 (28.0)	41 (8.3)	36 (7.3)	493 (100)	0.67 (0.58; 0.75) 1.00	
Insomnia medicine							U = 8491.500; *p* < 0.001
Yes	12 (17.6)	21 (30.9)	17 (25.0)	18 (26.5)	68 (100)	1.60 (1.34; 1.86) 2.00	
No	301 (58.6)	141 (27.4)	42 (8.2)	30 (5.8)	514 (100)	0.61 (0.54; 0.69) 0.00	
Anxiety medicine							U = 14,213.500; *p* < 0.001
Yes	29 (29)	25 (25.0)	20 (20.0)	26 (26.0)	100 (100)	1.43 (1.20; 1.66) 1.00	
No	284 (58.9)	137 (28.4)	39 (8.1)	22 (4.6)	482 (100)	0.58 (0.51; 0.66) 0.00	
Depression medicine							U = 11,426.000; *p* < 0.000
Yes	20 (20.6)	29 (29.9)	19 (19.6)	29 (29.9)	97 (100)	1.59 (1.36; 1.81) 1.00	
No	293 (60.4)	133 (27.4)	40 (8.2)	19 (3.9)	485 (100)	0.56 (0.48; 0.63) 0.00	

## Data Availability

The datasets generated during and/or analysed during the current study are available from the corresponding author on reasonable request.

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
