# Peer review of "Mental Health Evaluation in Community Pharmacies—A Cross-Sectional Study"

_pharmacy, 2024, doi:10.3390/pharmacy12030089_

Round 1

Reviewer 1 Report (Previous Reviewer 3)

Comments and Suggestions for Authors

I read with interest the revised version of the paper. The changes were well incorporated. 

Just some minor comments: 

1. A paragraph on the chapter of conclusions is needed.

2. Please remove white page (Page 14)

3. Bibliography after 46 have different format (no tab in the beggining)

4. Regarding significance tests, symbols and "." and "," were not used consistently. Diferent decimals are being used along those results. 

5. Table 5 - Format of Insomnia medicine and Anxiety medicine shoudl be reviewed and consistently formatted as the entire table. 

6. Table 3 - Median should be consistently presented as one or two decimals. Both options are present.  

Author Response

Dear Reviewer,

Thank you for your comments and suggestions. Below is our response, point by point:

  1. We correct it in the manuscript
  2. We think it might be a formatting problem. We can't see any blank pages in our Word document.
  3. We correct it in the manuscript
  4. We correct it in the manuscript
  5. We correct it in the manuscript
  6. We correct it in the manuscript

Once again, thank you very much.

Reviewer 2 Report (Previous Reviewer 1)

Comments and Suggestions for Authors

This study describes the use of community pharmacies in southern Portugal to screen people for symptoms of anxiety and depression using two types of questionnaires.  One questionnaire screened for anxiety and the other was for depression.  Questionnaires were scored numerically in a Likert-type scale.  Results show undiagnosed anxiety in about 18% and depression in 12% with about 40% of cases scoring as moderate to severe.  Data is based on self-reporting by participants.

Limitations are adequately discussed.  Results are carefully placed into context with similar studies in Europe. 

In this revision, the authors have clarified more the gap between the pharmacy’s screening for symptoms of depression and anxiety and obtaining a diagnosis; although the authors provided no point by point response to the first review. 

The author’s have also re-written some of the sentence fragments in the first version. 

There has been a 3-year period since the study began and data was gather during the severe acute respiratory syndrome coronavirus 2 period.  How are authors insuring current relevance of data?  This question was not addressed, yet since this current study began in 2000, post-COVID disease depression has become recognized.  The authors referenced at least three papers documenting COVID associated depression, but did not elaborate on any COVID effect on their data and results.  Could this explain in part why the prevalence of anxiety and depression in this study was higher than other studies?

Author Response

Dear Reviewer,

Thank you for your comments and suggestions. Below is our response, point by point:

  • We think there may have been a mistake or misunderstanding. When we resubmit the article, we submit a document with the responses to all the reviewers, point by point. We can resubmit this response if necessary.
  • The authors acknowledge that some time has elapsed since the data were collected, but we believe that the data are still relevant given the national mental health panorama and the reduced evidence of data collected in community pharmacy practice. It is important to remember that according to data from 2019 (before the COVID-19 pandemic), Portugal was already considered the country in the European Union with the highest burden of mental health problems, with anxiety disorders being the most common, followed by depressive disorders (2). The situation has worsened with the COVID-19 pandemic. A national survey conducted between May and July 2020 found that 27% of the population had symptoms of moderate to severe anxiety, and about the same percentage (26%) had symptoms of depression and post-traumatic stress disorder (7). In addition, according to the country's health profile for 2023, the COVID-19 pandemic has led to an increase in demand for specialised mental health services in Portugal, putting significant pressure on the health system. Although the number of psychiatrists increased by 18% between 2016 and 2021, barriers to access persist, mainly due to an insufficient supply of clinical staff and a lack of standardised referral criteria, which in turn exacerbates waiting lists for specialised mental health services (2). In this context, the authors believe that the data are still relevant, as anxiety and depression problems are unlikely to have resolved or disappeared.
  • Yes, the reviewer is absolutely correct. In fact, we didn't consider the possible contribution of depression in the context of COVID-19 in our analysis. However, we are planning to write a paper with a more detailed analysis of our data in the context of COVID-19.

Once again, thank you very much.

Reviewer 3 Report (New Reviewer)

Comments and Suggestions for Authors

The manuscript aims to presents the results of an observational cross-sectional study performed on patients in Portuguese pharmacies. The manuscript presents the data concerning the occurrence of anxiety and depression and shows major factors influencing predisposing patients to these disorders. The main strength of the manuscript are as follows: a study performed in real-life setting, the use of validated tools, i.e. GAD-7 and PHQ-9 questionnaires. Data presentation is clear and consistent, the article is well written. The research aim is clearly stated. The authors clearly explained the factors influencing the incidence of analyzed disorders in various patient subpopulations. The conclusions are consistent with the presented evidence. The authors identified some knowledge gaps that could be addressed by further studies.

I would raise some points that could be better addressed by authors:

1)     Although the authors clearly presented the research aim, it’s unclear what was the research gap in relation to pharmacy practice? The aim of assessing prevalence and connecting it to sociodemographic and medical factors has been achieved. However many studies trying to do a similar thing have already been published and factors influencing prevalence of these disorders have previously been identified. When it comes to the screening undiagnosed patients and monitoring treatment efficiency in diagnosed ones, one would like to know what happened following the completion of the survey. In the previously published studies that authors cited, i.e. https://doi.org/10.1007/s11096-019-00897-1 and DOI: 10.9740/mhc.2017.05.101 the pharmacists performed some interventions, i.e. concerning medication adherence, counselling or referral that could benefit the patient.

In the manuscript the authors claim that their aim was to propose a mean to monitor the selected conditions in diagnosed patients, but this would require some consistent, regular and well planned interventions. It seems that the study protocol focused more on screening rather than monitoring, as no information was provided on regular screening. It could be interesting for the reader of Pharmacy to know whether such intervention could provide a meaningful and beneficial support to disease management. Moreover, when it comes to patients participating in the study, what was the benefit of it, were they informed referred or addressed in any other relevant way? These issues should be addressed by authors.

2)     The section 3 should include the response rate. Did every participant that agreed to participate fill out the questionnaire completely?

3)     Section 2 should list exclusion criteria in addition to inclusion criteria. It should be also stated whether the pharmacies included in the study represented both rural and municipal areas.   

4)     How was the group of patients taking medicines for pain identified? Were they patients taking any pain relieving medicines, or those prescribed by a physician, or prescription-only? Could authors specify it?

5)     Literature cited in the manuscript is relevant to the paper. However some papers are old. This can be partially attributed to using well established research tools. However, some references, cited only vaguely could be replaced with newer literature, e.g. items 23, 24.

Author Response

Dear Reviewer,

Thank you for your comments and suggestions. Below is our response, point by point:

  1. We would like to make it clear that our study aimed to show that the involvement of community pharmacies in the detection and monitoring of anxiety and depression can make an important contribution to the early detection of new cases and the monitoring of those already diagnosed. However, we would like to emphasise that our study was only observational, so we did not carry out any interventions as part of the study. We believe that colleagues in the pharmacies where the study was conducted acted according to their internal procedures for the situations.
    We would like to add that we intend to conduct a study in the future to assess the impact of pharmacist intervention in this area.
  2. In the design of the study, we took into account the recording of refusals. As the referee points out, there were in fact some refusals and some incomplete questionnaires. The data presented in the study do not indicate the refusal rate or the number of incomplete questionnaires, as these figures are not relevant to the study proposal.
  3. We have included the exclusion criteria in the methodology. As for the location of pharmacies, we didn't include this variable because the majority of the data was collected from urban pharmacies.
  4. Pain medication use was self-reported by the person, regardless of medical prescription
  5. We have updated reference 23.
    However, with regard to reference 24, we didn't find any more recent studies in the same context. In fact, there are several recent studies that estimate the prevalence of anxiety in the United States, but they were conducted on specific populations, such as immigrants and veterans. It seems to us that substitution would not be appropriate. There is also a recent study (Journal of Affective Disorders 336 (2023) 81-91) that estimates the overall prevalence of anxiety in the United States at 44%, but this study was based on an analysis of the national database and was not conducted in primary care settings. 

In this context, we decided not to change the reference.

Once again, thank you very much.

This manuscript is a resubmission of an earlier submission. The following is a list of the peer review reports and author responses from that submission.

Round 1

Reviewer 1 Report

Comments and Suggestions for Authors

This study describes the use of community pharmacies in southern Portugal to screen people for symptoms of anxiety and depression using two types of questionnaires.  One questionnaire screened for anxiety and the other was for depression.  Questionnaires were scored numerically in a Likert-type scale.  Results show undiagnosed anxiety in about 18% and depression in 12% with about 40% of cases scoring as moderate to severe.  Data is based on self-reporting by participants.

Results provide useful information confirming anxiety and depression symptoms more common in females, and lower levels of education but somewhat surprisingly not altered by per ownership. 

Data also show specific correlations with pain medications, but did not assess with any type of internal control maintenance drugs such as statins or antilipidemic drugs to ensure the correlation is not with all medications implicating the pharmacy patient population.

Limitations are adequately discussed.  Results are carefully placed into context with similar studies in Europe. 

The main area for improvement is the need to identify a link between pharmacy questionnaire screening and therapy.  The middle steps after screening were not explored.  These would be diagnosis, drug selection if needed and payer acceptance.  The Discussion mentions medication management, promoting adherence, but this is not directly relevant to the study since there is not even a definitive diagnosis let alone pharmacotherapy.  How does initial screening tie into a role for pharmacy and pharmacists?

Other considerations: 

There has been a 3-year period since the study began.  How are authors insuring current relevance of data?

Authors state that “The presence of pets had no influence on anxiety and depression symptoms” but I did not see the pet question listed in the PHQ-9 table.

Of the undiagnosed participants who denied taking anxiety medications, was this verified by checking the participant’s pharmacy profiles?  Anxiety or depression medications would likely have been filled by the pharmacy.  This would help flag false negatives.

The Abstract’s 2nd sentence “To screen undiagnosed people for symptoms of anxiety and depression, and to assess the control of these conditions in diagnosed patients.” Is not a sentence.

“Observational, cross-sectional study of a sample of community pharmacy users” is not a sentence.

Comments on the Quality of English Language

The Abstract’s 2nd sentence “To screen undiagnosed people for symptoms of anxiety and depression, and to assess the control of these conditions in diagnosed patients.” Is not a sentence.

“Observational, cross-sectional study of a sample of community pharmacy users” is not a sentence.

Reviewer 2 Report

Comments and Suggestions for Authors

Thank you for the opportunity to review this paper. Here are some major concerns:

-first of all the introduction of the article is to short. You must include more data about healthcare in Portugal, the way pharmacist interract with patients and what kind of problems they encounter.

- the procedure of completing the questionnaire is not clear and needs more explaining. 

- the disscusion part is too long. Please rephrase it having in consideration more bibliographic references. 

Minor concerns:

- rephrase the introduction

- line 158 - 161 ? explain this paragraph - the study aim is to identify patients depression or pharmacists depression? because this one states that the study identifies the need for pharmacists to intervene in the early identification of patients with depression.

- delete paragraph: 251-253

- rephrase the disscusion part.

Reviewer 3 Report

Comments and Suggestions for Authors

I read with interest the paper titled "Mental health evaluation in community pharmacy – A cross-sectional study"

- Background - What means a rate of 18.4% of mental illness? Is it good or bad compared to other European/Worldwide countries?

- I was a bit confused after reading the first lines of the introduction. First you talk about OECD, then Europe, then US. I suggest going from broader info and then focus in your population?

- Why the focus only on community pharmacists? In the entire Europe you have other professionals, as Pharmacy Techncians and Assistants, in the frontline of Community Pharmacies, with direct contact with public. Aren't they also in good position for that? What is the status in Portugal?

The text is too much focused on pharmacists. Also, in the manuscript, pharmacies were invited, not pharmacists, which could be a bias of saying that a specific professional category is better positioned. Please describe.

Methods

- What is the non-probablistic method used? All patients during the 11 months of the study were invited to participate? Please describe. 

- Why the data is so old? 3 years has passed since the collection. Is it still significant? The data was collected during a period of COVID (stated in the background that represents higher scores of anxiety and depression). 

- Also, during the collection time, a 2 months shut down have ocurred. Please describe how you collect data during this period.

- Which are the background of the pharmacy employees that collected the data? Please add this information in the manuscript. 

- What happened to people with more than 10 in the GAD-7 score? Were them referenced from the general pratitioners? Since, further evaluation is recommended when the score is 10 or greater, this is a question that I would like to be clarified. 

- A pilot study was carried out. How many participants? Were them include in the final sample? What was changed after the pilot?

- Study was approved by ethics committee of the University of Algarve (CEUAlg Pn°32a)/2021). Seems the study was approved in 2021, however the collection of data started in 2020. Please clarify the date when the study was approved. Seems that data collection started without the study approved, is that correct?

3. Results

- If one of the inclusion criteria was age, how was age not available for 51 participants. They should be removed from the study. 

- When report, please be consistent in one or two decimals (eg: (SD=16.62).)

- Figure 1 - Undiagnosed are more than diagnosed? This is an interesting result. Please provide the comparison tests p-values and when different, please discuss further. 

- Table 2 shows significance, but I cannot understand from which is refered. For eg. between education level, a chi-square was used, but it was not the relevant test to be used. You can either compare between the qualitative scores (and the test to be used must be ANOVA or KW, depending of assumptions). You can also compare groups within eacg qualitative score, which will be a good add (and then again ANOVA or KW is the test to be considered, after testing the assumptions (with Levene test and Kolmogorov-Smirnov or Shapiro-Wilk)

- Table 3, 4 and 5 are titled as correlation. No correlations were provided. I suggest strongly review the statistical section.

Conclusion

- The overall discussion and conclusions focus more on opinions from the authors and discussion of non-related results, than in the results of the study itself. It appears that the authors have the desire to highlight the role of pharmacists in some anxiety/depression evaluation. 

The aim of the study was "screen undiagnosed people for symptoms of anxiety and depression and to assess the control of these conditions in diagnosed patients. In addition, we aimed to assess the prevalence of symptoms based on sociodemographic factors, chronic illness and isolation.". 

a) A comparison of undiagnosed/diagnosed must be provided and was not. 

b) An evaluation of the disease control in the diagnosed people must be provided and was not. 

c) I suggest to present the items in the both scales and compare the groups. It was not done yet. 

Comments on the Quality of English Language

The English must be extensively improved. I suggest proofreading services for that or the text must be reviewed by a native speaker. 

Reviewer 4 Report

Comments and Suggestions for Authors

The major concerns are as follows.

Introduction: The introduction requires additional information on the state of the art. Are there any similar studies conducted globally?

Methods: There is a lack of information regarding the validity of the questionnaire used. Additionally, details about how the pharmacies were selected, trained in sampling the respondents, and the process of when and how the questionnaire was answered are missing. Furthermore, specific aims and methods of data analysis are not adequately provided.

Results: There is an excess of information in the text that should be tabulated for reader-friendliness (refer to page 4). Tables 3-5 need better descriptions.

Discussion: The rationale in the discussion is unclear. Are you discussing the prevalence of depression and anxiety in the countries, or are you focusing on the ability of pharmacists to screen for these conditions? The discussion also touches on the sensitivity and specificity of the screening instrument, making it even more unclear what the primary focus and main message of your study are.